# Clofarabine Preconditioning followed by Allogeneic Transplant Using TBI and Post-Transplant Cyclophosphamide for Relapsed Refractory Leukemia

**DOI:** 10.3390/ijms25020957

**Published:** 2024-01-12

**Authors:** Seema Naik, Kevin Rakszawski, Hong Zheng, David Claxton, Kentaro Minagawa, Shin Mineishi

**Affiliations:** Department of Medicine, Penn State Cancer Institute, 500 University Dr. Hershey, Hershey, PA 17033, USA; krakszawski@pennstatehealth.psu.edu (K.R.); hzheng@pennstatehealth.psu.edu (H.Z.); dclaxton@pennstatehealth.psu.edu (D.C.); kminagawa@pennstatehealth.psu.edu (K.M.); smineishi@pennstatehealth.psu.edu (S.M.)

**Keywords:** AML: acute myeloid leukemia, Allo-HSCT: allogeneic transplant, Clo: clofarabine, PTCy: post-transplant cyclophosphamide

## Abstract

Acute myeloid leukemia patients with induction failure or relapsed refractory disease have minimal chance of achieving remission with subsequent treatments. Several trials have shown the feasibility of clofarabine-based conditioning in allogeneic stem cell transplants (allo-HSCT) for non-remission AML patients. Pre-transplant conditioning with clofarabine followed by reduced-intensity allo-HSCT has also demonstrated a potential benefit in those patients with human leukocyte antigen (HLA)-identical donors, but it is not commonly used in haploidentical and mismatched transplants. In this case report, we describe our experience of seven cases of non-remission AML who received clofarabine preconditioning followed by an allo-HSCT with PTCy. The 2-year overall survival and disease-free survival was 83.3% (95% confidence interval (CI): 27.3–97.9%) and 85.7% (95% CI: 33.4–97.9%). Median days of neutrophil and platelet recovery were 16 (range of 13–23) and 28 (range of 17–75), respectively. The cumulative incidence of grade II-IV acute graft-versus-host disease (GVHD) at day 100 and chronic GVHD at 1-year showed 28.6% (95% CI: 8–74.2%) and 28.6% (95% CI: 3–63.9%), respectively. The two-year relapse rate was 14.3% (95% CI: 2.14–66.6%). One-year GVHD-free relapse-free survival (GFRS) at 1-year was 71.4% (95% CI: 25.8–92%). Our patients showed successful outcomes with clofarabine preconditioning to reduce the leukemic burden at the pre-transplant period followed by PTCy to reduce GVHD resulting in lower relapsed rate and better GFRS in these patients.

## 1. Introduction

For patients with acute myeloid leukemia (AML) who fail to achieve complete remission (CR) or those in relapse after first CR (CR1), allogeneic hematopoietic cell transplantation (Allo-HCT) can produce leukemia control and extended survival. Some reports suggest that immediate transplantation is the best strategy for primary induction failure (PIF) or at first relapse (Rel1) if a donor is quickly available [1,2]. For the patients who do not achieve remission or those with relapsed refractory disease, the probability of achieving remission with subsequent induction chemotherapies is minimal. Allogeneic hematopoietic stem cell transplantation (allo-HSCT) is the only curative option for these patients. A high post-transplant relapse rate and transplant-related mortality often preclude them from proceeding to transplant. Thus, AML not in remission at the time of allo-HSCT remains a substantial unmet need in current practice, particularly for patients without a human leukocyte antigen (HLA)-matched donor.

Clofarabine is a second-generation purine nucleoside analog with substantial single-agent activity in adult patients with relapsed and refractory AML [1]. It is also known to be active in combination with other cytotoxic agents [2]. Therefore, the anti-leukemic activity of clofarabine has attracted attention as an effective treatment for patients with relapsed and refractory leukemia. In addition, clofarabine has a strong immunosuppressive effect, and several trials have shown the feasibility of conditioning with a clofarabine-based regimen [3,4]. Previously, we used a conditioning regimen of clofarabine and busulfan for 4 days (CloBu4) in allo-HSCT for non-remission AML, and day 30 complete remission rates were 90–100% [5,6]. Of note, these patients were transplanted only from HLA-matched donors.

Preconditioning is another strategy to improve the outcome for high-risk leukemia. The sequential treatment with chemotherapy and reduced-intensity conditioning for allo-HSCT has shown a potential benefit in relapsed and refractory patients [7,8], where the induction chemotherapy was given just before conditioning and patients received allo-HSCT at nadir. More recently, clofarabine has been used for preconditioning [9,10,11,12,13]. However, clofarabine is less frequently used in haploidentical and mismatched transplants. We previously reported the strategy where we used clofarabine as a preconditioning chemotherapy with a post-transplant cyclophosphamide (PTCy) post-conditioning regimen [14]. In this report, we describe our experience of seven cases of non-remission AML (shown in Table 1 and Table 2) who received clofarabine preconditioning followed by allo-HSCT with PTCy.

**Methods:** The patient received clofarabine as preconditioning chemotherapy followed by conditioning regimen at Penn State Cancer Institute, Milton Hershey Medical Center, between July 2017 and July 2020. The patients received post-transplant cyclophosphamide (PTCy) on day 3 and day 4 of the post-transplant conditioning regimen [14]. The patients signed the informed consent prior to transplant, and these seven cases were retrospectively analyzed in accordance with Penn State University Institutional Review Board (IRB)-approved protocol.

Transplant conditioning regimens were as follows, as shown in Figure 1. Regimen A with clofarabine followed by fludarabine and busulfan (FluBu2)/TBI/PTCy was used for younger AML patients. Regimen B with clofarabine followed by fludarabine and cyclophosphamide (FluCy)/TBI/PTCy was used for younger AML patients. Regimen C with clofarabine followed by cyclophosphamide (Cy)/TBI/PTCy was used for patients with CML blast crisis.

The patients received tacrolimus (FK 506) and mycophenolate mofetil (MMF) starting from day 5, according to standard haploidentical or mismatched transplant regimens. GCSF was administered daily from day +5 until neutrophil engraftment. Transfusion support and electrolyte replacements were administered according to standard institutional guidelines. The patients were followed daily during hospitalization until neutrophil engraftment. After hospital discharge, patient had follow-up bone marrow biopsies and peripheral blood chimerism studies on day 30, day 100, day 180, and then yearly post-transplant.

## 2. Case Presentation

Case 1 was a 40-year-old AML patient with fms-like tyrosine kinase 3 internal tandem duplication (Flt3-ITD), resulting in three induction failures. Sorafenib was the only tyrosine kinase inhibitor (TKI) available at that time. It was effective only for a short period, with 45% blasts in the bone marrow. The patient received clofarabine preconditioning, followed by a haploidentical transplant with a fludarabine (Flu)/Bu3/total body irradiation (TBI) conditioning regimen with PTCy. Bone marrow on day 17 post-transplant indicated complete remission (CR), but minimal residual disease (MRD) was positive for Flt3-ITD. He started on the compassionate use of gilteritinib as post-transplant maintenance. He achieved a molecular remission and is still in molecular remission now, almost 5 years after allo-HSCT.

Case 2 was a 34-year-old patient initially diagnosed with AML positive for ASXL1. He received 7 + 3 induction chemotherapy, but bone marrow at recovery showed residual AML with blasts of 28.5%. He received reinduction with cladribine/cytarabine/granulocyte colony-stimulating factor/mitoxantrone (CLAG-M) [15], but bone marrow still showed relapsed refractory AML with 60% blasts. He underwent a haploidentical allo-HSCT from his brother with clofarabine for 5 days before conditioning with Flu/Bu3 with TBI 400 cGy followed by PTCy. His transplant course was complicated by neutropenic fever. Overall, he tolerated the procedure well. He received azacytidine maintenance because he did not have a targetable mutation. He was then readmitted for diarrhea which proved to be acute graft-versus-host disease (GVHD). His diarrhea improved with steroids and infliximab. He had recurrent flares of gut GVHD treated with steroids and budesonide with improvement in symptoms. He has been tapered off steroids and is currently only on sirolimus and continues in remission for more than 5 and a half years.

Case 3 was a 50-year-old patient diagnosed with chronic myelogenous leukemia (CML) lymphoid blast crisis initially treated with dasatinib, but who was then switched to nilotinib due to resistance (ABL E225K) mutation followed by hyper CVAD salvage regimen. He achieved a CR with positive MRD following induction chemotherapy. He underwent matched unrelated donor allo-HSCT with Flu/Melphalan (Mel) conditioning. His post-transplant course was complicated by acute gut GVHD requiring prolonged steroids. He relapsed one and a half years later with 87% blasts and was found to have thoracic T4 epidural mass causing spinal cord compression. The pathology was consistent with CML lymphoid blast crisis. He received ponatinib, prednisone, and weekly vincristine and achieved CR, but bcr-abl PCR was still positive at the international scale (IS) at 0.00053%. He received chimeric antigen receptor (CAR)-T cells (CD19-41 BBL) and then was maintained on ponatinib. At 6 months post-CAR-T, BCR-ABL p210 transcript was (0.003%), and bone marrow again showed relapsed ALL with BCR-ABL E255K mutation. He initiated ponatinib, venetoclax, and decitabine but developed transaminitis resulting in cessation of chemotherapy. His CSF cytology was positive for 24% blasts six weeks before transplant. He was treated with intrathecal methotrexate. His pre-transplant marrow showed 86% blasts. He then underwent a haploidentical allo-HSCT from his daughter with clofarabine (40 mg/m^2^ for 4 days), with 3 days of rest, followed by modified Cy/TBI conditioning (Cy 20 mg/kg on day −3, TBI 800 cGy over 2 days in four fractions, and PTCy 50 mg/kg × 2 on days 3 and 4). Post-transplant bone marrow revealed a normocellular marrow with normal cytogenetics and negative BCR/ABL with a five-log reduction in the tumor burden. He continues on ponatinib and venetoclax for post-transplant maintenance and has remained in molecular CR for 2 years and 11 months.

Case 4 was a 36-year-old patient diagnosed with high-risk myelodysplastic syndrome (MDS) with progressive pancytopenia with AML a year later. Her cytogenetic study showed high-risk features with 20q deletion, 5q deletion, and 11q23 abnormalities. She received an allo-HSCT from a matched sibling donor with the Bu/Cy regimen. She achieved CR and was doing well post-allo-HSCT, with mild liver chronic GVHD, which was resolved eventually. However, she developed pancytopenia 2 and a half years after allo-HSCT. Bone marrow showed MDS with excess blast (EB) 2 with 13% blasts with 20 q deletion, trisomy 9, and U2AF1 mutation. She received a cycle of azacytidine but still had a persistent 28% blasts concerning refractory leukemia. She was started on decitabine for 10 days with venetoclax but developed Enterobacter sepsis, pseudomonas bacteremia, and multifocal fungal pneumonia. Her counts were slow to recover, but bone marrow showed 40% blasts with the persistent clonal disease. She then received clofarabine for 5 days followed by a modified CyTBI with PTCy, the same regimen as case 3. Her day +100 bone marrow biopsy showed no signs of AML with normal cytogenetics. She was doing well post-transplant but developed fungal pneumonia precluding the initiation of maintenance therapy. Eventually, her day 180 bone marrow showed relapse with the same complex karyotype, and she died of disease progression.

Case 5 was a 73-year-old patient with a diagnosis of AML who received decitabine and venetoclax for induction chemotherapy with a good response. He received six cycles of treatment when he relapsed. He received a reinduction chemotherapy with CLAG followed by azacytidine with venetoclax, but failed to attain CR. He received a clofarabine preconditioning followed by Flu/Cy/TBI/PTCy allo-HSCT (Hopkins non-myeloablative regimen [16]). He developed cytomegalovirus viremia and Strep viridians bacteremia at the time of engraftment, which resolved with antiviral agents and antibiotics. His day 30 and 100 bone marrows were consistent with CR. He started maintenance therapy with azacytidine and has been in CR for over 3 years.

Case 6 was a 59-year-old patient with AML, initially treated with the 7 + 3 induction chemotherapy regimen and achieved CR1, which was followed by high-dose Ara-C consolidation. He experienced a relapse with leukemic cutis 5 months later. He then received reinduction chemotherapy (CLAG-M) and achieved CR2. He relapsed again with leukemia cutis and failed decitabine/venetoclax. He then received CLAG-M with a response, but then developed progressive leukemia cutis which was treated with Ara-C 100 mg/m^2^ for 3 days for cytoreduction awaiting transplant. He then received clofarabine preconditioning (30 mg/m^2^ for 5 days) and a modified CyTBI regimen for a matched unrelated donor (MUD) HSCT with PTCy. He developed neutropenic fevers and mucositis complicating his early recovery phase post-transplant. His post-transplant bone marrow was consistent with CR. He received no systemic maintenance therapy post-transplant but received prophylactic electron beam radiation to the involved skin area. He has remained in CR for two years post-transplant.

Case 7 was a 36-year-old patient with GATA-2-deficient AML background MDS with monosomy 7 karyotype and Emberger’s syndrome [17]. Her initial bone marrow biopsy showed AML with dysmegakaryopoiesis and background myelodysplasia and 25% blasts positive for CD33 with monosomy 7. The molecular panel showed a GATA–2E 219 RFS x3 mutation and was negative for other targetable mutations. She received azacitidine × 3 cycles followed by decitabine for 5 days and venetoclax reinduction chemotherapy with persistent MRD. The patient received another cycle of decitabine and venetoclax chemotherapy, but her bone marrow showed persistent 30% blasts. She then received clofarabine followed by a Flu/Bu2/TBI regimen with PTCy. Her day 30, 100, and 180 bone marrows were in remission. The patient now has resolved chronic limited skin GVHD and has been in remission for 24 months post-transplant.

Overall, the 2-year overall survival and disease-free survival of our cases was 83.3% (95% confidence interval (CI): 27.3–97.9%) and 85.7% (95% CI: 33.4–97.9%) (as shown in Figure 2 and Table 3). Median days of neutrophil and platelet recovery were 16 (range of 13–23) and 28 (range of 17–75), respectively. The cumulative incidence of grade II-IV acute GVHD at day 100 and chronic GVHD at 1 year showed 28.6% (95% CI: 8–74.2%) and 28.6% (95% CI: 3–63.9%), respectively. The two-year relapse rate was 14.3% (95% CI: 2.14–66.6%). One patient relapsed and died as she could not receive her post-transplant maintenance therapy due to ongoing comorbidities, including fungal pneumonia. One-year GVHD free relapse-free survival (GFRS) at one year was 71.4% (95% CI 25.8–92%) (as shown in Figure 2 and Table 3).

## 3. Discussion

Clofarabine exhibits efficacy in hematologic malignancies such as acute lymphoid leukemia (ALL), AML, and MDS [1,18,19]. In the phase I/II trial, clofarabine was given in combination with a myeloablative dose of Bu (CloBu4), as a pre-transplant conditioning regimen for patients with refractory hematologic malignancies [5]. All patients engrafted rapidly, and this combination showed promising antitumor activity in these very high-risk patients with 48% one-year overall survival and 20% long-term disease-free survival [5]. A multicenter study with this regimen showed similar results where 2-year overall and event-free survivals were 26% and 20%, respectively [6]. CloBu4 has been tested only for HLA-matched donors but not yet for haploidentical and mismatched unrelated donor transplant. Chevallier and his colleagues demonstrated that a clofarabine-based Hopkins-style PTCy regimen provides safety and a better outcome for AML patients, including 40% of active disease with 18-month overall and disease-free survivals of 72% and 63.8%, respectively [20]. This may indicate that the clofarabine regimen from those donors benefits high-risk leukemia when combined with PTCy.

In treating very aggressive leukemias, another approach was taken, as shown in a European study, to start transplant conditioning immediately after a few days of preconditioning chemotherapy followed by days of rest before the start of transplant conditioning. In early articles, the chemotherapy used upfront was fludarabine, high-dose AraC, and amsacrine (FLAMSA), which was considered very effective in aggressive leukemias [7,8]. Locke et al. reported prospective phase II study results of 29 patients given clofarabine 30 mg/m^2^ for 5 days followed immediately by allo-HSCT conditioning while at the cytopenic nadir [21]. Toxicities were acceptable, with transient hyperbilirubinemia (48%) and grade 3–4 infections (10%). Post-allo-HSCT, non-relapse mortality at day 180 was 7% (95% CI: 1–21%), relapse was 29% (95% CI: 13–46%), and overall survival was 71% (95% CI: 51–85%), comparing favorably with published data for high-risk patients. Richardson and colleagues conducted 11 cases of clofarabine preconditioning in patients with high-risk, refractory acute leukemia or advanced MDS [9]. Six patients received CyTBI conditioning, and five received various reduced-intensity conditioning. The patients tolerated conditioning well with one-year overall survival of 48% in this highly aggressive AML population, and one-year transplant-related mortality was 8%. Other groups showed similar results [10,11,12,13]. Thus, clofarabine used as a bridge to allo-HSCT to reduce disease burden was well tolerated in high-risk refractory AML patients with acceptable toxicities. Tischer and her group reported clofarabine (30 mg/m^2^ for 5 days) as preconditioning and Flu/Cy/Mel/TBI/PTCy for haploidentical transplant in high-risk AML patients; estimated overall survival and relapse-free survival at 1 year from haploidentical allo-HSCT were 56 and 39%, respectively [22]. Our experience also showed an excellent 2-year overall survival (83.3%) and disease-free survival (85.7%) for those high-risk AML patients from haploidentical, matched, and mismatched unrelated donors. Collectively, a preconditioning strategy using clofarabine may improve the outcome of high-risk AML patients when used in combination with PTCy, regardless of the specific donor type.

We initially used FluBu3 as the transplant conditioning with moderate to severe toxicity, even in relatively young patients. We then decided to use FluBu2 thereafter to decrease early post-transplant toxicity. Alternatively, we used Flu/Cy/TBI 800cGy with PTCy for a few additional patients younger than 60 years, and our 73-year-old AML (case 6) received clofarabine conditioning with Fly/Cy/TBI 200cGy followed by PTCy. Toxicities, including infectious disease and mucositis, were manageable, and acute and chronic GVHD incidence was low (28.6% in both). No transplant-related mortality was seen. Tischer’s group also reported that the rate of acute GVHD grade II-IV was 22%, and chronic GVHD occurred in five patients (28%), with non-relapse mortality after 1 year being 23%. Considering the patient’s age and comorbidities and adjusting the conditioning regimen at an appropriate intensity that does not increase toxicity can also be critical in achieving optimal results.

## 4. Conclusions

New evidence has been emerging for the use of maintenance therapy in AML with or without targetable mutations [23]. We used allo-HSCT as a central platform with pre-transplant and post-transplant strategies linked together with the goal of long-term remission for refractory leukemia patients. Our successful cases received maintenance therapy and continue in remission post-transplant. Therefore, it is crucial to avoid GVHD and infections that interfere with early maintenance therapy, especially when treating high-risk AML. PTCy may help in this regard.

We experienced successful outcomes in patients with relapsed and refractory hematological malignancies treated with clofarabine containing preconditioning followed by allo-HSCT with PTCy. “Preconditioning” may help reduce the leukemic burden pre-transplant, and PTCy may help reduce GVHD in these relapsed/refractory leukemia patients. This strategy may improve the outcomes of relapsed/refractory leukemia patients.

## Figures and Tables

**Figure 1 ijms-25-00957-f001:**
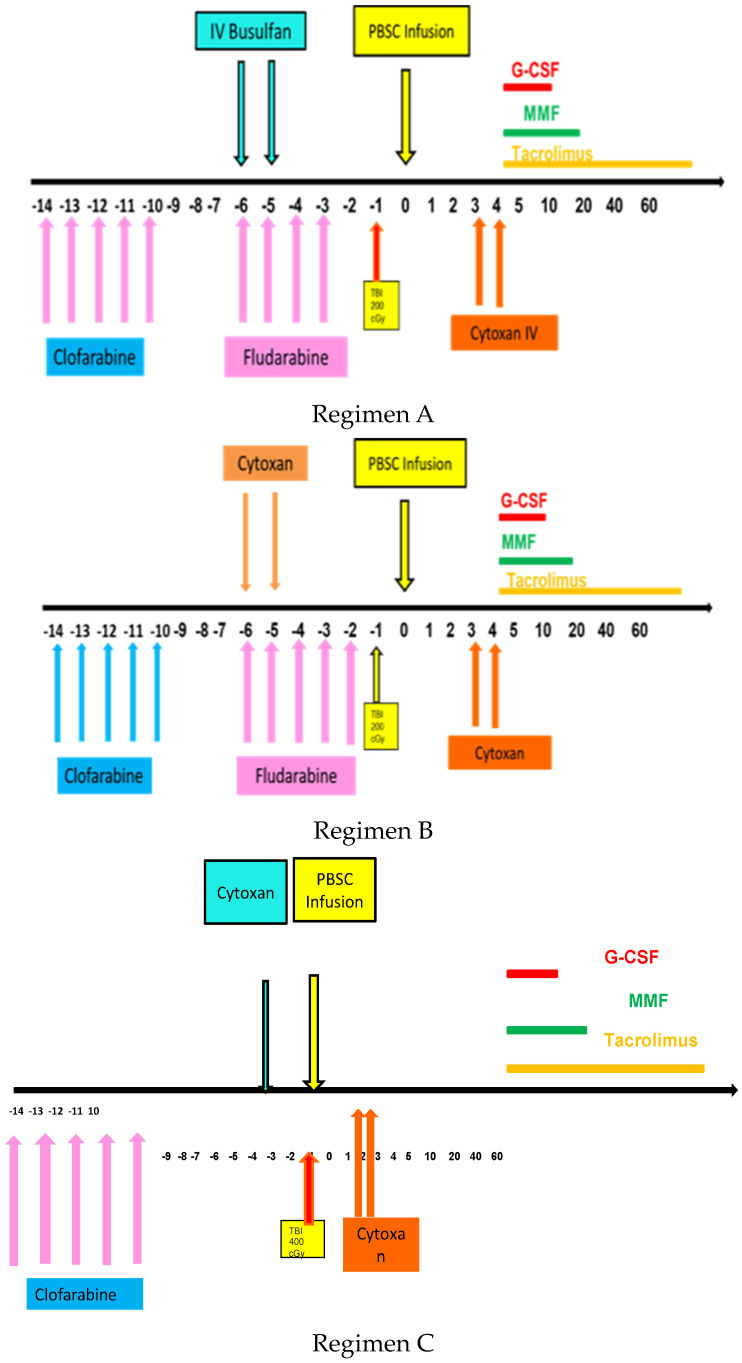
Schema for clofarabine based regimens followed by transplant with TBI and PTCy.

**Figure 2 ijms-25-00957-f002:**
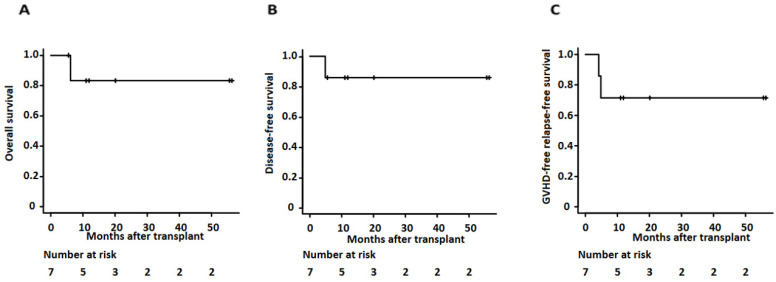
Kaplan–Meier survival curves for (**A**) overall survival, (**B**) disease-free survival, and (**C**) GVHD-free relapse-free survival for patients undergoing clofarabine preconditioning followed by allogeneic transplant using TBI and PTCy.

**Table 1 ijms-25-00957-t001:** Characteristics of patients undergoing clofarabine followed by allogeneic transplant.

#	Disease/Age	Age	Disease Status at Transplant	Conditioning Regimen	Donor	Neutrophil Recovery	Complications	Acute GVHD	Chronic GVHD
1	AML	40	PIF	Clo/Flu/Bu3/TBI (200cGy) /PTCy	Haplo Brother	13	Engraftment syndrome	None	Moderate
2	AML	31	PIF	Clo/Flu/Bu3/TBI (400cGy)/ PTCy	Haplo Sister	15	Neutropenic fevers, mucositis	None	Severe
3	CML	50	BC	Clo/Cy/TBI (800cGy)/ PTCy	Haplo Daughter	20	Neutropenic fevers, mucositis	None	None
4	AML	36	1st Rel	Clo/Cy/TBI (800cGy)/ PTCy	Haplo Sister	19	PRES	Grade II	None
5	AML	72	PIF	Clo/Flu/CY/TBI (200cGy)/ PTCy	MUD 8/8	23	AKI, Strep. Viridans bacteremia	None	Mild
6	AML	59	2nd Rel	Clo/Cy/TBI (800cGy)/ PTCy	MUD 8/8	16	Neutropenic fevers, mucositis	Grade II	None
7	AML	36	PIF	Clo/Flu/Bu2/TBI (200cGy)/ PTCy	MUD 7/8	16	Mucositis	Grade I	Moderate

Abbreviations: GVHD: graft-versus-host disease; AML: acute myeloid leukemia; PIF: primary induction failure; Clo: clofarabine; Flu: fludarabine; Bu: busulfan; TBI: total body irradiation; PTCy: post-transplant cyclophosphamide; Haplo: haploidentical transplant; CR: complete remission; CML: chronic myelogenous leukemia; BC: blast crisis; Rel: relapse; PRES: posterior reversible encephalopathy syndrome; MUD: matched unrelated donor; AKI: acute kidney injury.

**Table 2 ijms-25-00957-t002:** Blast percentage/mutational status and post-transplant maintenance.

#	Blast %	Mutation	Karyotype	Day +30	Maintenance	Current Status
1	45%	FLT3	Normal	CR/pos MRD	Gilteritinib	Alive in CR
2	60%	ASXL1	Normal	CR	Azacytidine	Alive in CR
3	86%	ABL E225K)	T(9/22)	CR	Ponatinib Venetoclax	Alive in CR
4	40%	20q del, 5q del, 11q23	Complex	CR	None	Relapse/died at D+180
5	50%	None	-Y	CR	Azacytidine	Alive in CR
6	5% Leukemia Cutus	ASXL1	Normal	CR	Electron beam therapy	Alive in CR
7	30%	GATA–2	Normal	CR	None	Alive in CR

**Table 3 ijms-25-00957-t003:** Survival outcomes post-clofarabine followed by transplant for relapsed refractory AML.

Outcome	%	95% CI
2-year OS	83.3%	27.3–97.9%
2-year DFS	85.7%	33.4–97.9%
Neutrophil recovery	16	13–23
Platelet recovery	28	17–75
Gr II-IV acute GVHD	28.6%	8–74.2%
Chronic GVHD at 1 yr	28.6%	3–63.9%
2-yr relapse rate	14.3%	2.14–66.6%
1-yr GFRS	71.4%	25.8–92%

## Data Availability

Supplementary data are available in our database reported to CIBMTR.

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
