# Peer review of "Clofarabine Preconditioning followed by Allogeneic Transplant Using TBI and Post-Transplant Cyclophosphamide for Relapsed Refractory Leukemia"

_ijms, 2024, doi:10.3390/ijms25020957_

Round 1
Reviewer 1 Report
Comments and Suggestions for Authors
The work of Naik et al. describes a case series of 7 patients with relapsed/refractory myeloid malignancies preconditioned with clofarabine with immediate HSCT with post-transplant cyclophosphamide. Outcome of those patients is quite impressive with a 2-year overall survival and
disease-free survival of 83.3% and 85.7% respectively. Toxicity seems acceptable. Given the small number of patients publication seems justified as data in this setting are sparse and the paper is written in a case report format.
Some issues:
1. Table 1 needs profound revision, mutations and karyotype should be separated. Mutational status should be addressed for all patients if available as well as karyotype and blast count prior clofarabine. Patients should be classified according to WHO 2022 and ICC criteria if possible. Response status for each patient before HSCT should be included. Maybe 2 tables for better visibility would be more appropriate
2. Pp3/4 Outcome data should be separated in a chapter “Outcome”
3. Pp 4; ll 154-160 refers to Figure 1 and 2. I found only 1 Figure (2) in the supplemental data. This should be clarified
4. Figure(s) and Table(s) should be integrated in the manuscript
Author Response
Some issues:
- Table 1 needs profound revision, mutations and karyotype should be separated. Mutational status should be addressed for all patients if available as well as karyotype and blast count prior clofarabine. Patients should be classified according to WHO 2022 and ICC criteria if possible. Response status for each patient before HSCT should be included. Maybe 2 tables for better visibility would be more appropriate: will revise table 1.
- Pp3/4 Outcome data should be separated in a chapter “Outcome”Will separate outcomes data.
- Pp 4; ll 154-160 refers to Figure 1 and 2. I found only 1 Figure (2) in the supplemental data. This should be clarified. Will add both figures 1 and 2 .
- Figure(s) and Table(s) should be integrated in the manuscript: Will add figured and tables in th emnuscript.
Reviewer 2 Report
Comments and Suggestions for Authors
The present paper reports 7 separate cases of relapsed refractory leukemia that were treated with allogeneic-HSCT, where Clofarabine was used for pretransplant conditioning. The reported positive outcomes have been ascribed to the synergy of cytotoxic and immunosuppresive effects of Clofarabine, that helps reducing leukemic burden prior to transplantation and reduces the probability for GVHD. The current paper represents an introduction to a future prospective clinical trial.
The provided results are clear and valuable. Some minor points:
- Please re-write more precisely the first paragraph in the Introduction, particularly the first sentence, and add references.
- Wherever applicable, use as precise terms as possible, for example “induction chemotherapy” instead of just “induction” or “pre-transplant conditioning” instead of “preconditioning”.
-Please transfer Table 1 to the results section. It is difficult to track and compare the cases without tabular representation.
-Please arrange all data in percents from the Abstract into another table, and put into the Results section.
- Abbreviations used in Abstract should be defined in Abstract, and, if further used, they should be defined once more in the main text.
Author Response
- Please re-write more precisely the first paragraph in the Introduction, particularly the first sentence, and add references. Introduction first sentence rewritten wiht reference.
- Wherever applicable, use as precise terms as possible, for example “induction chemotherapy” instead of just “induction” or “pre-transplant conditioning” instead of “preconditioning”. Correction made to ass “induction chemotherapy” instead of just “induction” or “pre-transplant conditioning” instead of “preconditioning”.
-Please transfer Table 1 to the results section. It is difficult to track and compare the cases without tabular representation. Table 1 transferred to "results" secion.
-Please arrange all data in percents from the Abstract into another table, and put into the Results section. Another table added to arrange data form abstract to "results" section.
- Abbreviations used in Abstract should be defined in Abstract, and, if further used, they should be defined once more in the main text. "Abbreviations " defined again in the main text.